# Two-Tier Care Pathways for Liver Fibrosis Associated to Non-Alcoholic Fatty Liver Disease in HIV Mono-Infected Patients

**DOI:** 10.3390/jpm12020282

**Published:** 2022-02-15

**Authors:** Giada Sebastiani, Jovana Milic, Adriana Cervo, Sahar Saeed, Thomas Krahn, Dana Kablawi, Al Shaima Al Hinai, Bertrand Lebouché, Philip Wong, Marc Deschenes, Claudia Gioè, Antonio Cascio, Giovanni Mazzola, Giovanni Guaraldi

**Affiliations:** 1Chronic Viral Illness Service, McGill University Health Centre, Montréal, QC H4A 3J1, Canada; adriana.cervo@gmail.com (A.C.); thomas.krahn@mail.mcgill.ca (T.K.); bertrand.lebouche@mcgill.ca (B.L.); 2Division of Gastroenterology and Hepatology, Royal Victoria Hospital, McGill University Health Centre, Montréal, QC H4A 3J1, Canada; dana.kablawi@mail.mcgill.ca (D.K.); pw0867@gmail.com (P.W.); marc.deschenes@muhc.mcgill.ca (M.D.); 3Centre for Outcomes Research and Evaluation, Research Institute of the McGill University Health Centre, Montréal, QC H4A 3J1, Canada; 4Department of Surgical, Medical, Dental and Morphological Sciences, University of Modena and Reggio Emilia, 41121 Modena, Italy; jovana.milic@gmail.com (J.M.); giovanni.guaraldi@unimore.it (G.G.); 5Department of Infectious Diseases, Azienda Ospedaliero-Universitaria di Modena, 41124 Modena, Italy; 6Department of Internal Medicine—Infectious Disease, Center for Dissemination and Implementation, Institute for Public Health, School of Medicine, Washington University, St. Louis, MO 63110-1010, USA; zssaeed@gmail.com; 7Division of Experimental Medicine, McGill University, Montréal, QC H4A 3J1, Canada; al.alhinai@mail.mcgill.ca; 8Department of Family Medicine, McGill University, Montréal, QC H3S 1Z1, Canada; 9Infectious and Tropical Disease Unit, AOU Policlinico “P. Giaccone”, 90127 Palermo, Italy; claudiagioe86@gmail.com (C.G.); antonio.cascio03@unipa.it (A.C.); 10Department of Health Promotion, Mother and Child Care, Internal Medicine and Medical Specialties-University of Palermo, 90127 Palermo, Italy; 11Infectious Diseases Unit, S. Elia Hospital, 93100 Caltanissetta, Italy; gnni.mazzola@gmail.com

**Keywords:** transient elastography, controlled attenuation parameter, serum fibrosis biomarkers, APRI, FIB-4

## Abstract

(1) Background: Developing strategies to identify significant liver fibrosis in people with HIV (PWH) is crucial to prevent complications of non-alcoholic fatty liver disease (NAFLD). We aim to investigate if five simple serum biomarkers applied to PWH can optimize a care pathway to identify significant liver fibrosis defined by transient elastography (TE). (2) Methods: A two-tier fibrosis pathway was applied to three prospective cohorts of PWH undergoing TE with CAP. NAFLD was diagnosed as a controlled attenuation parameter ≥ 248 dB/m. Five simple fibrosis biomarkers (FIB-4 < 1.3, BARD score 0–1, NAFLD fibrosis score < −1.455, AST:ALT ratio < 0.8 and APRI < 0.5) were applied as first-tiers to exclude significant liver fibrosis. We determined the decrease in referral for TE that would have occurred based on biomarker assessment and the discordance between low simple fibrosis biomarkers and high TE (≥7.1 kPa), indicating significant liver fibrosis. (3) Results: Of the 1749 consecutive PWH, 15.1% had significant liver fibrosis by TE and 39.1% had NAFLD. The application of the fibrosis biomarkers as first tiers would have resulted in a decrease in TE referrals between 24.9% (BARD score) and 86.3% (APRI). The lowest discordance rate was with NAFLD fibrosis score (8.5%). After adjustments, BMI (odds ratio (OR) 1.12, 95% CI: 1.08–1.17) and triglycerides (OR 1.26, 95% CI: 1.11–1.44) were independent predictors of discordance for APRI < 0.5 and TE ≥ 7.1. The performance of the two-tier pathways was similar in PWH with and without NAFLD. (4) Conclusions: Implementing a two-tier pathway could save a substantial proportion up of TE examinations, reducing costs and helping resource optimization in HIV care. Patients with metabolic risk factors for NAFLD and low fibrosis biomarker may still be considered for TE referral.

## 1. Introduction

Non-alcoholic fatty liver disease (NAFLD) represents a global epidemic, with a prevalence at 25.24% [1]. If left unmanaged, NAFLD could lead to non-alcoholic steatohepatitis (NASH), liver fibrosis accumulation, cirrhosis and end-stage liver complications. NAFLD seems frequent in people with HIV (PWH), with a prevalence ranging from 13 to 65% [2,3,4,5,6,7]. This burden is likely driven by a multifactorial complex pathogenesis, resulting from lifelong use of ART, past exposure to hepatotoxic d-drugs (didanosine and stavudine), persistent immune activation and HIV-related inflammation, and very prevalent dysmetabolic conditions [8,9]. NAFLD may not only be more prevalent in PWH than in the HIV-uninfected population, but also more severe: both NASH and significant liver fibrosis seem at least twice more frequent in HIV mono-infected patients than in the general population [10,11,12,13,14].

NAFLD is often asymptomatic until patients develop hepatic decompensation, with significant morbidity and mortality and related socio-economic burden [15]. The stage of liver fibrosis represents the main prognostic factor of all-cause and liver-related mortality in NAFLD [16]. There is a need for personalized medicine and implementation of strategies to identify those who have significant liver fibrosis and are at risk of poor outcomes. The guidelines from the European AIDS Clinical Society (EACS) recommend the case-finding of significant liver fibrosis in PWH with metabolic conditions or persistent elevated transaminases [17]. These recommendations are in line with other at-risk populations for NAFLD-related liver fibrosis, such as patients with type-2 diabetes [18,19]. Transient elastography (TE) with controlled attenuation parameter (CAP) is a feasible and accurate tool to assess both hepatic steatosis and NAFLD-associated liver fibrosis in PWH [20]. The utilization of this specialist test facilitates the implementation of therapeutic interventions and surveillance for complications associated with advanced liver fibrosis, including screening for hepatocellular carcinoma (HCC) and esophageal varices. However, diagnosing liver fibrosis is a challenge due to the large number of PWH at risk for NAFLD, as well as the additional resources of delivering TE with CAP, which is often not readily accessible in clinics practicing HIV care. Clinical pathways have been proposed in HIV-uninfected NAFLD to screen for liver fibrosis in at-risk populations and reduce the need for specialist tests [21,22,23]. In these models, readily available and inexpensive fibrosis biomarkers with high negative predictive value (NPV) are used as first-tier tests, while more specialized tests, such as TE with CAP, are second-tier tests reserved to cases in which fibrosis cannot be excluded by the simple biomarker [21,23]. Among the simple fibrosis biomarkers potentially suitable as first-tier tests, the aspartate aminotransferase (AST): alanine aminotransferase (ALT) ratio, the fibrosis-4 (FIB-4), the AST-to-Platelet Ratio Index (APRI), the NAFLD fibrosis score and the BARD score have been sufficiently validated for the diagnosis of significant liver fibrosis, as well as for their prognostic value in predicting liver-related events and mortality [24,25]. The performance characteristics of the clinical pathways employing simple fibrosis biomarkers and TE with CAP for significant liver fibrosis in PWH, who are at risk for NAFLD, are unknown. 

The aim of this study is to investigate if five simple serum biomarkers applied to PWH can optimize a two-tier care pathway for the identification of significant liver fibrosis as defined by TE. Specifically, we evaluate the reduction in the TE referral rate and related costs that would have occurred based on biomarker assessment, and the discordance rate between TE and simple fibrosis biomarkers. Finally, we determine the factors associated with the discordance between serum biomarkers and TE.

## 2. Materials and Methods

We conducted a retrospective cross-sectional study from the LIVEr disease in HIV (LIVEHIV), Modena HIV Metabolic Clinic (MHMC) and Liver pathologies in HIV in Palermo (LHIVPA) Cohorts [10,26,27]. The LIVEHIV Cohort is a prospective routine screening program for NAFLD and liver fibrosis established in September 2013 at McGill University Health Centre (MUHC) in Montreal, Canada. HIV-infected patients undergo screening for NAFLD and liver fibrosis by TE with CAP [10]. The MHMC Cohort was initiated in 2004 in Modena, Italy, to assess longitudinal metabolic changes among PWH through annual comprehensive assessment and TE with CAP [27]. The LHIVPA Cohort was initiated in 2011 at the Infectious Diseases Outpatient Clinic of the University Hospital in Palermo, Italy. Metabolic assessment through physical and biochemical parameters and TE with CAP is conducted annually [28]. We included all consecutive patients with HIV infection (documented by positive enzyme-linked immunosorbent assay (ELISA) with Western blot confirmation) aged ≥ 18 years with availability of TE with CAP and relevant clinical and biochemical parameters. Exclusion criteria were: (i) positivity for HCV antibody or hepatitis B surface antigen; (ii) evidence of other liver disease; (iii) significant alcohol intake, defined as more than 30g/day in men and more than 20g/day in women [19]; (iv) history of HCC or liver transplantation; (v) contraindications (pregnancy, pacemaker insertion) and failure or unreliable measurement of TE examination with CAP; and (vi) missing liver transaminases or platelets.

We included patients with available data within 3 months from the TE examination, namely demographic information, HIV and medications history, BMI, liver serum biomarkers, lipid profile, hematological and immuno-virological parameters. Type-2 diabetes mellitus was defined as a hemoglobin glycosylated of 6.5% or greater, or as previously diagnosed by an endocrinologist/treating physician. Hypertriglyceridemia was defined as triglycerides ≥1.7 mmol/L. Elevated liver transaminases were defined as either ALT > 45 IU/L or AST > 35 IU/L.

TE examinations were performed on a 4 h fasting patient by a maximum of two experienced operators at each site (>500 examinations before the study) [29]. The standard M probe was used in all patients. The XL probe was used in the case of failure with M probe and if BMI > 30 Kg/m^2^ [30]. The following criteria were applied to define the result of liver stiffness measurement (LSM) as reliable: at least 10 validated measures and an interquartile range <30% of the median. A cut-off of LSM by TE ≥ 7.1 kPa was used to define significant liver fibrosis, corresponding to stage F2-F4 out of 4 by the histologic Kleiner staging system [31,32]. NAFLD was defined as CAP ≥ 248 dB/m [7,33].

Five simple fibrosis biomarkers were adopted as first-tier tests. Previously proposed validated cut-offs were used to rule-out significant liver fibrosis and subsequent need for the second-tier test TE. The five biomarkers included:-AST:ALT ratio [34]. A cut-off < 0.8 was used to exclude significant liver fibrosis [24,35].-AST-to-Platelet Ratio Index (APRI) = (AST/upper limit of normal)/platelets × 100. A cut-off < 0.5 was used to exclude significant liver fibrosis [36].-FIB-4 = (age × AST)/platelets × √ALT. A cut-off < 1.30 was used to exclude significant liver fibrosis [24,37].-BARD = BMI ≥ 28 kg/m^2^ (yes = 1, no = 0) + AST:ALT ratio ≥ 0.8 (yes = 2, no = 0) + type-2 diabetes (yes = 1, no = 0). A cut-off < 2 was used to exclude significant liver fibrosis [38].-NAFLD fibrosis score = −1.675 + (0.037 × age) + (0.094 × BMI) + (1.13 × diabetes (yes = 1, no = 0)) + (0.99 × AST:ALT ratio) − (0.013 × platelets) − (0.66 × albumin (g/dL)). A cut off < −1.455 was used to exclude significant liver fibrosis [39].

The primary study outcome was the decrease in TE referral rate that would have occurred when using simple fibrosis biomarkers as first-tier tests in a two-tier pathway (Figure 1A), compared to the use of TE in all patients (Figure 1B). Secondary outcomes include: (a) rate of discordance between high TE (≥7.1 kPa) and each among low simple fibrosis biomarker (FIB-4 < 1.3, BARD score 0–1, NAFLD fibrosis score < −1.455, AST:ALT ratio < 0.8 and APRI < 0.5); (b) predictors of discordance between high TE (≥7.1 kPa) and low simple fibrosis biomarkers; (c) cost analysis of two-tier pathways compared to the use of TE in all patients.

Potential direct cost savings were estimated using Canadian data. As previously reported [40,41], costs of non-invasive tests in Canadian dollars were as follows: FIB-4 and APRI: CAD 17; NAFLD fibrosis score: CAD 22; AST:ALT ratio and BARD score: CAD 10; TE: CAD 125. Total direct cost saved was computed by subtracting the direct cost of serum biomarker to the cost saved for the TE examination.

We compared the characteristics of the participants by significant liver fibrosis status by TE using Student’s t-test for continuous variables and Pearson’s χ^2^ for categorical variables. The two-tier pathways were modelled using one of the simple fibrosis biomarkers as first-tier and TE as the second-tier test. The reduction in the referral rate for TE tests, along with the discordance rate, was computed. A subgroup analysis was conducted in patients with NAFLD diagnosed by CAP. Predictors of discordance between low simple fibrosis biomarkers and TE ≥ 7.1 kPa were determined using unadjusted and adjusted logistic regression models and reported as adjusted odds ratios (aOR) with 95% confidence interval (CI). All adjusted regression models included covariates that were determined a priori to be clinically important, based on the previous literature, or those with *p*-value < 0.05 in univariable analysis. Final models were adjusted for age, sex, BMI, diabetes, triglycerides, and CD4 cell count. A complete case analysis was used for the multivariable models and the percentage of missing data were less than 15%, unless otherwise indicated. All tests were two-tailed and with a significance level of α = 0.05. Statistical analyses were performed using STATA 15 (STATA Corp. LP, College Station, TX, USA).

## 3. Results

After applying the inclusion and exclusion criteria, a total of 1749 PWH were included (Figure 2), of whom 582 were from the LIVEHIV cohort, 649 from the MHMC cohort and 518 from the LHIVPA cohort. 

Mean age was 50.2 years (standard deviation (SD) 10.4) and 74.5% were males. A total of 1483 (84.8%) PWH included were of white ethnicity. The mean CD4 cell count was 702.9 cells/μL (SD 318.3) and mean duration of HIV infection was 15.5 years (SD 9.8). Most of the PWH included in this study had well controlled HIV: the proportion of participants with a CD4 cell count <200 cells/mm^3^ and <350 cells/mm^3^ was 2.6% and 11.3%, respectively. All patients were on ART, including 1497 (85.6%) on nucleoside/nucleotide reverse transcriptase inhibitors, 773 (44.2%) on nonnucleoside reverse transcriptase inhibitors, 932 (53.3%) on protease inhibitors and 619 (35.4%) on integrase inhibitors. Overall, 132 (7.5%) had a history of exposure to didanosine. The distribution of risk factors for NAFLD was as follows: 44.8% PWH were overweight, 34% had diabetes and 12.8% had elevated liver transaminases. In the whole cohort, 264 (15.1%) had significant liver fibrosis by LSM. Table 1 reports the characteristics of the study population with univariable analysis by significant liver fibrosis status, as per LSM. When compared to patients without significant liver fibrosis, PWH with fibrosis were older, had higher BMI and longer time since HIV diagnosis. They also had a higher prevalence of diabetes. Finally, PWH with significant liver fibrosis had higher ALT, AST, total cholesterol, triglycerides, albumin, CAP, while having lower platelets. They also presented a tendency for lower CD4 cell count. All the serum fibrosis biomarkers were higher in PWH with significant liver fibrosis by LSM, except for the AST:ALT ratio.

### 3.1. Application of the Two-Tier Pathway

The application of the simple fibrosis biomarkers as first-tier tests would have resulted in a significant proportion of PWH identified as not at risk for significant liver fibrosis, with consequent decrease in TE referrals ranging between 24.9% (BARD score) and 86.3% (APRI) (Table 2). The discordance rate between low fibrosis biomarker and high LSM ranged between 8.5% (NAFLD fibrosis score) and 19.5% (AST: ALT ratio). 

Since APRI and FIB-4 yielded the highest decrease in TE referrals, we conducted a multivariable analysis of independent predictors of discordance between low APRI (<0.5) or FIB-4 (<1.3) and high TE (≥7.1). After adjustments, high BMI and triglycerides were independent predictors of discordance for APRI and FIB-4 (Table 3).

In the cost analysis, APRI showed the highest total direct cost saved (CAD 9088 per 100 PWH), followed by FIB-4 (CAD 6175 per 100 PWH), NAFLD fibrosis score, AST:ALT ratio and BARD score (Table 2).

### 3.2. Effect of NAFLD

The overall prevalence of NAFLD in our cohort of consecutive PWH was 39.1%. The prevalence of significant liver fibrosis by LSM was higher in patients with NAFLD (24.3%) compared to those without NAFLD (9.2%) (*p* < 0.001). Patients with NAFLD had higher rates of discordance low fibrosis biomarker and high LSM by TE (Figure 3). Conversely, there was no difference in decrease in TE referral according to NAFLD status (data not shown). 

Table 4 shows the characteristics of the 264 PWH with significant liver fibrosis by NAFLD status. Those PWH with significant fibrosis but without NAFLD had lower BMI, CD4 cell count, platelets, total cholesterol and triglycerides. They also had higher HDL cholesterol, AST: ALT ratio, FIB-4 and APRI. Finally, PWH with significant liver fibrosis without NAFLD was in a higher proportion in females and in the parameter of exposure to didanosine.

## 4. Discussion

A two-tier care pathway using simple fibrosis biomarkers as first-tier tests in a large population of consecutive PWH could reduce the referral rate to tertiary care for TE examination up to 86%, with a significant reduction in terms of cost and use of healthcare resources. We also found that APRI and FIB-4 reduced the most referral rates and costs, while the NAFLD fibrosis score had the lowest discordance rate with LSM. Finally, our study indicated that discordance rates between FIB-4 or APRI and LSM are associated with overweight and hypertriglyceridemia, as such TE examination may still be considered in PWH even in the case of low fibrosis biomarker.

With the increasing life expectancy of PWH thanks to the widespread use of ART, NAFLD is now a major contributor to the burden of liver disease. Studies across different countries place the prevalence of the disease between 28 and 72.1% [20]. NAFLD seems not only more frequent in the setting of HIV infection, but also more severe. NASH has been reported in up to 63.6% of HIV mono-infected patients with chronically elevated ALT and in 10% of those attending a routine screening program [13,20,42,43,44]. PWH with NAFLD have more advanced liver disease and higher rates of NASH compared to age/sex-matched HIV-negative controls [11]. The prevalence of NAFLD-associated significant liver fibrosis ranges between 13.8 and 28.6% [20,43,45]. Similarly, in our study we found a prevalence of significant liver fibrosis of 15.1%. Unfortunately, NAFLD lacks specific signs or symptoms, and between 67% and 83% of PWH with NAFLD have normal ALT [7,46]. As such, a significant proportion of NAFLD patients receive a diagnosis at the first episode of hepatic decompensation due to liver cirrhosis [47]. Similarly to other at-risk populations for NAFLD, such as patients with type-2 diabetes [18,19], recent EACS guidelines recommend the staging of liver fibrosis in PWH at risk for NAFLD, particularly in case of metabolic comorbidities [17]. Due to the high prevalence of NAFLD in PWH, it is unfeasible to perform tertiary care specialized tests for fibrosis staging in all patients, such as TE. Two-tier care pathways have been recently proposed in populations at risk for NAFLD, such as patients with type-2 diabetes or those with unexplained elevation of liver transaminases. In primary care, these pathways involve the use of simple fibrosis biomarkers to stratify patients at risk for NAFLD who have a higher likelihood of having significant liver fibrosis. These readily available simple fibrosis biomarkers used as first-tier tests have low cost and high NPV. Davyduke et al. reported that a FIB-4 first care pathway used in 565 patients at risk for NAFLD from primary care could save 87% of specialist tests, including TE and liver biopsy [21]. Similarly, a U.K. study of 3012 patients with risk factors for NAFLD found that a pathway using FIB-4 followed by more specialized fibrosis tests could save 88% unnecessary referrals to tertiary care [23]. With up to 86% saved TE examinations, our study achieved similar results in the specific setting of HIV mono-infection. Indeed, biopsy-proven studies in PWH showed a similarly high NPV of simple fibrosis biomarkers to exclude significant liver fibrosis. In 49 PWH, Lemoine et al. reported a NPV of 95% for FIB-4 < 1.30, 92% for APRI < 0.5 and 90% for NAFLD fibrosis score < −1.455 compared to liver histology [31]. In the two-tier pathway we used, only patients with a high fibrosis biomarker would be referred to tertiary care for TE. Our study found that APRI and FIB-4 had the highest rates of decrease in TE referral to tertiary care, 86.3% and 60%, respectively. These figures would also result in a reduction in costs: compared to the use of TE in all PWH, the APRI first and FIB-4 first pathways would save CAD 9088 and CAD 6175 per 100 PWH, respectively. As for NAFLD fibrosis score, although the decrease in TE referrals and costs was inferior, the discordance rate between this biomarker and LSM by TE was the lowest among the simple fibrosis biomarkers investigated. Conversely, our study showed that the simple fibrosis biomarkers AST:ALT ratio and BARD score perform less than APRI, FIB-4 and NAFLD fibrosis score in a two-tier pathway in PWH. These results are in agreement with a report of 743 PWH without viral hepatitis, where TE and simple liver fibrosis scores showed a significant degree of discordance, proposing to limit that use of single diagnostic tools in routine clinical practice [48]. The multivariable analysis found that high BMI and triglycerides were independent predictors associated with a discordance between low APRI (<0.5) or FIB-4 (<1.3) and high LSM (≥7.1). This finding could be used in clinical practice, suggesting that PWH with these metabolic comorbidities may be considered for TE referral even in the case of low APRI or FIB-4, which in fact do not contain any metabolic variable in their formula. This was further strengthened by the fact that PWH with NAFLD diagnosed by CAP had higher rates of discordance for the simple fibrosis biomarkers. Giving that both BMI and triglyceride levels are dynamic variables, they should be reassessed at each time point when the two-tier care pathway is applied. This is particularly relevant for BMI, given that recent studies suggest weight gain overtime with specific ART regimens, such as integrase inhibitors [49]. Moreover, dietary patterns, such as the Mediterranean diet, can influence both BMI and triglycerides in NAFLD, thus pointing out the need for the longitudinal re-assessment of these variables [50]. The EACS guidelines recommend periodical re-assessment of fibrosis every 2–5 years, depending on clinical risk [17]. 

The overall prevalence of NAFLD in our at-risk cohort was 39.1%, a figure which is in line with previous reports in PWH cohorts and with a meta-analysis [4,10,27]. In PWH with NAFLD, we found a higher prevalence of significant liver fibrosis by TE (24.3%) compared to those without NAFLD. However, the prevalence of significant liver fibrosis in PWH without NAFLD was still 9.2%, pointing out that HIV mono-infected patients can develop significant liver fibrosis in the absence of NAFLD. These liver fibrosis cases may be due to risk factors for liver disease other than the metabolic ones and that are unique to HIV infection, including chronic inflammation and exposure to ART [20,51]. Indeed, patients with significant liver fibrosis and without NAFLD had a lower CD4 cell count and a higher proportion of exposure to didanosine, while metabolic comorbidities and high BMI were less frequent. Previous reports suggested low CD4 cell counts to be associated with liver fibrosis [52]. Even in the group of PWH without NAFLD, APRI and FIB-4 outperformed the other simple fibrosis biomarkers as first-tier tests. Our findings underline the need for considering liver disease in HIV mono-infected patients beyond NAFLD, especially in the presence of the unexplained elevation of liver transaminases.

We wish to acknowledge several limitations of this study. First, it was conducted at tertiary-care referral centers, and so the prevalence of significant liver fibrosis may be higher than in primary care settings. Secondly, liver biopsy, the gold standard to diagnose NAFLD and stage liver fibrosis, was unavailable. However, the EACS guidelines recommend non-invasive tests as initial diagnostic tools, as liver biopsy is not recommended for screening and routine clinical use [17].

## 5. Conclusions

A two-tier care pathway using fibrosis biomarkers as first-tiers could save up to 86.3% of TE examinations and care costs, helping personalized medicine and resource optimization in HIV clinics and low-resource settings. Similar to NAFLD in the HIV-uninfected population, APRI and FIB-4 seem the most promising first-tier tests. This pathway could help the case-finding of at-risk PWH who will benefit from specialist referral for secondary fibrosis assessment, such as TE examination. The identification of PWH at a high risk for liver fibrosis is of paramount importance considering that there is no pharmacologic treatment for NAFLD-associated liver fibrosis and that PWH are excluded from most ongoing global clinical trials for NASH [53]. Future studies should focus on the longitudinal and long-term effects of the large-scale application of these models of care. 

## Figures and Tables

**Figure 1 jpm-12-00282-f001:**
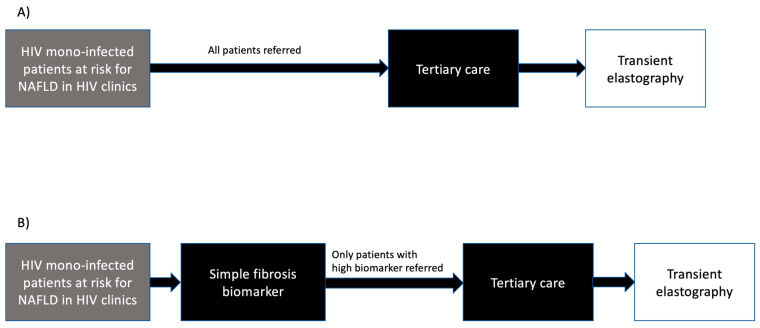
Schematic representation of the two scenarios of referral to tertiary care. The first scenario (**A**) assumes that patients have simple a fibrosis biomarker tested in primary care and those at risk for significant liver fibrosis are referred to tertiary care to undergo transient elastography. The second scenario (**B**) assumes that all patients are referred to tertiary care to undergo transient elastography. High simple fibrosis biomarker was defined as follows: FIB-4 ≥ 1.3, BARD score 2–4, NAFLD fibrosis score ≥ −1.455, AST:ALT ratio ≥ 0.8 or APRI ≥ 0.5.

**Figure 2 jpm-12-00282-f002:**
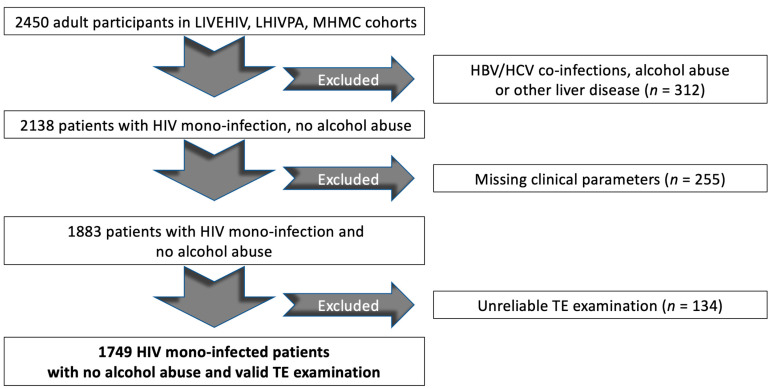
Flow chart displaying the selection of study participants in the cohort. Abbreviations: LIVEHIV, LIVEr disease in HIV; MHMC, Modena HIV Metabolic Clinic; LHIVPA, Liver pathologies and HIV in Palermo; HBV, hepatitis B virus; HCV, hepatitis C virus; HIV, human immunodeficiency virus; TE, transient elastography.

**Figure 3 jpm-12-00282-f003:**
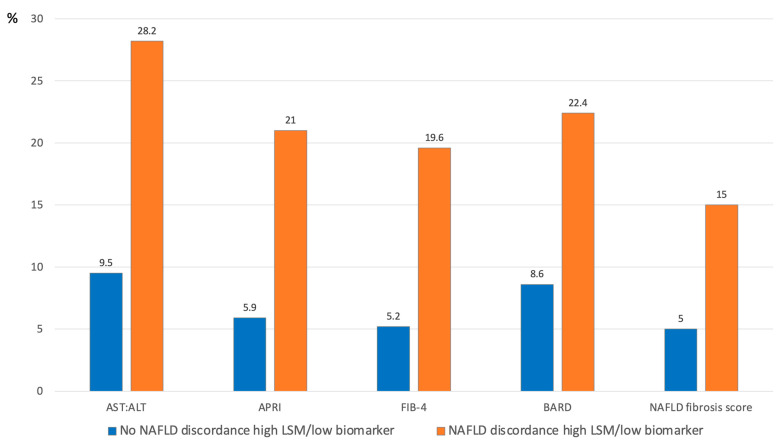
Proportion of discordance between low fibrosis biomarker (first-tier test) and high LSM by TE (second-tier test) by NAFLD status.

**Table 1 jpm-12-00282-t001:** Characteristics of PWH by significant liver fibrosis status as determined by LSM with transient elastography (*n* = 1749).

	LSM ≥ 7.1 kPa (*n* = 264)	LSM < 7.1 kPa (*n* = 1485)	*p*
Age (years)	53.1 (9.4)	49.7 (10.5)	<0.001
Male sex (%)	203 (76.9)	1100 (74.1)	0.333
Ethnicity (%)			
White/Caucasian	231 (87.5)	1250 (84.2)	0.167
Black non-Hispanic	25 (9.5)	168 (11.3)	
Diabetes (%)	110 (41.7)	484 (32.6)	<0.001
BMI (Kg/m^2^) °	27.2 (5.3)	24.7 (4.0)	<0.001
Time since HIV diagnosis (years)	18.9 (10.3)	14.9 (9.6)	<0.001
Undetectable HIV viral load (<40 copies/mL) (%)	203 (77.0)	1099 (74.0)	0.322
CD4 cell count (cells/μL)	669.3 (347.4)	708.9 (312.6)	0.065
Current ART regimen (%)			
NRTIs	228 (86.4)	1268 (85.4)	0.678
NNRTIs	122 (46.2)	649 (43.7)	0.449
Protease inhibitors	148 (56.1)	768 (51.7)	0.192
Integrase inhibitors	83 (31.4)	538 (36.2)	0.134
Past exposure to didanosine (%)	23 (8.7)	107 (7.2)	0.390
ALT (IU/L)	35.5 (32.3)	24.5 (16.1)	<0.001
AST (IU/L)	31.7 (24.2)	23.1 (10.4)	<0.001
Platelets (10^9^ cells/L)	202.3 (74.3)	223.8 (62.3)	<0.001
Albumin (g/dL)	4.32 (0.55)	4.37 (0.40)	0.119
Triglycerides (mmol/L)	1.96 (1.76)	1.51 (1.06)	<0.001
Total cholesterol (mmol/L)	3.58 (1.89)	2.97 (2.23)	<0.001
HDL (mmol/L)	1.18 (0.38)	1.28 (0.39)	<0.001
CAP (dB/m)	269.2 (61.7)	230.8 (54.2)	<0.001
AST: ALT ratio	1.07 (0.65)	1.09 (0.44)	0.375
BARD score	2.20 (1.14)	2.01 (1.11)	0.015
NAFLD fibrosis score	−0.84 (1.55)	−1.70 (1.36)	<0.001
FIB-4	1.91 (2.00)	1.20 (0.67)	<0.001
APRI	0.60 (0.82)	0.33 (0.20)	<0.001

Notes: Continuous variables are expressed as mean (standard deviation) and categorical variables as number (%). The *p*-values refer to Student *t*-test or χ^2^ test between LSM ≥ 7.1 and LSM < 7.1 kPa. Data on BMI and BARD score were available for 1564 PWH (89.4%). Data on albumin were available for 1122 PWH (64.2%). Data on NAFLD fibrosis score were available for 1017 PWH (58.1%).

**Table 2 jpm-12-00282-t002:** Cost analysis of potential direct cost savings by the two-tier pathway, estimated using Canadian data.

	APRI	FIB-4	NAFLD Fibrosis Score	BARD Score	AST: ALT Ratio
Decrease in TE referral (%)	86.3	63.0	51.2	24.9	26.2
Discordance high LSM/ low biomarker (%)	11.7	11.0	8.5	15.4	19.5
Direct cost of serum biomarker per 100 PWH (CAD)	1700	1700	2200	1000	1000
TE cost saved per 100 PWH (CAD)	10,788	7875	6400	3113	3275
Total direct cost saved per 100 PWH (CAD)	9088	6175	4200	2113	2275

Notes: All dollar values are 2019 Canadian dollars.

**Table 3 jpm-12-00282-t003:** Multivariable analysis of predictors of discordance between high TE and low FIB-4 (*n* = 1101) or low APRI (*n* = 1510).

Variable	OR (95% CI)	aOR (95% CI)	*p*-Value
FIB-4			
Male sex (yes vs. no)	0.91 (0.60–1.39)	0.82 (0.51–1.31)	0.052
BMI (per Kg/m^2^)	1.14 (1.09–1.19)	1.14 (1.09–1.19)	<0.001
Diabetes (yes vs. no)	0.83 (0.56–1.28)	0.77 (0.48–1.23)	0.270
Triglycerides (per mmol/L)	1.29 (1.11–1.49)	1.23 (1.03–1.45)	0.019
CD4 cell count (per 100 cell/mL)	0.99 (0.99–1.00)	0.99 (0.99–1.00)	0.807
APRI			
Age (per 10 years)	1.27 (1.09–1.48)	1.19 (0.99–1.42)	0.054
Male sex (yes vs. no)	0.95 (0.67–1.35)	0.74 (0.50–1.09)	0.129
BMI (per Kg/m^2^)	1.12 (1.08–1.16)	1.12 (1.08–1.17)	<0.001
Diabetes (yes vs. no)	1.21 (0.87–1.68)	1.13 (0.78–1.64)	0.511
Triglycerides (per mmol/L)	1.32 (1.17–1.48)	1.26 (1.11–1.44)	0.001
CD4 cell count (per 100 cell/mL)	0.99 (0.99–1.00)	0.99 (0.99–1.00)	0.154

Notes: Odds ratios (OR) and 95% confidence interval (CI) are presented for each variable in the unadjusted and adjusted analysis. Continuous variables in the models were age, BMI, triglycerides, and CD4 cell count. Categorical variables in the models were male sex and diabetes.

**Table 4 jpm-12-00282-t004:** Characteristics of PWH with significant liver fibrosis by transient elastography by NAFLD status (*n* = 264).

	CAP ≥248 dB/m (n = 166)	CAP <248 dB/m (n = 98)	p-Value
Age (years)	53.6 (8.9)	52.3 (10.2)	0.293
Male sex (%)	135 (81.3)	68 (69.4)	0.026
Ethnicity (%)			
White/Caucasian	139 (83.7)	92 (93.9)	0.581
Black non-Hispanic	17 (10.2)	8 (8.2)	
Diabetes (%)	66 (60.0)	44 (40.0)	0.413
BMI (Kg/m^2^) °	28.7 (4.7)	24.4 (5.2)	<0.001
Time since HIV diagnosis (years)	18.5 (9.7)	19.5 (11.3)	0.435
Undetectable HIV viral load (<40 copies/mL) (%)	125 (75.3)	78 (79.6)	0.639
CD4 cell count (cells/μL)	711.3 (656.8)	599.3 (330.7)	0.012
Current ART regimen (%)			
NRTIs	141 (84.9)	87 (88.8)	0.380
NNRTIs	80 (48.3)	42 (43.9)	0.401
Protease inhibitors	101 (60.8)	62 (63.3)	0.696
Integrase inhibitors	48 (28.9)	35 (35.7)	0.250
Past exposure to didanosine (%)	6 (3.6)	17 (17.3)	<0.001
ALT (IU/L)	38.0 (34.9)	31.3 (27.0)	0.105
AST (IU/L)	30.7 (23.7)	33.3 (24.9)	0.410
Platelets (10^9^ cells/L)	214.2 (68.9)	182.1 (78.9)	<0.001
Albumin (g/L)	43.5 (5.4)	42.6 (5.7)	0.233
Triglycerides (mmol/L)	2.3 (2.1)	1.4 (0.8)	<0.001
Total cholesterol (mmol/L)	4.6 (1.1)	1.9 (1.7)	<0.001
HDL (mmol/L)	1.1 (0.4)	1.3 (0.4)	0.012
AST:ALT ratio	0.93 (0.34)	1.29 (0.93)	<0.001
BARD score	2.19 (1.14)	2.22 (1.14)	0.818
NAFLD fibrosis score	−0.95 (1.41)	−0.66 (1.75)	0.247
FIB-4	1.50 (1.11)	2.61 (2.83)	<0.001
APRI	0.50 (0.61)	0.77 (1.07)	0.008

Notes: Continuous variables are expressed as mean (standard deviation) and categorical variables as number (%). The *p*-values refer to Student *t*-test or χ^2^ test between CAP ≥ 248 dB/m and CAP < 248 dB/m. ° Data on BMI and BARD score were available for 237 PWH (89.8%). Data on albumin were available for 183 PWH (69.3%). Data on NAFLD fibrosis score were available for 159 PWH (60.2%).

## Data Availability

Data sharing statement: According to stipulations of the patient consent form signed by all study participants, the ethical restrictions imposed by our Institutional Ethics review boards (Institutional Ethics Review Board Biomedical B Research Ethics Board of the McGill University Health Centre), and the legal restrictions imposed by Canadian law regarding clinical trials, anonymized data are available upon request. Please send data access requests to Sheldon Levy, Biomedical B (BMB) Research Ethics Board (REB) Coordinator Centre for Applied Ethics, 5100, boul. de Maisonneuve Ouest, 5th floor, Office 576, Montréal, Québec, H4A 3T2, Canada.

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
