# Peer review of "Two-Tier Care Pathways for Liver Fibrosis Associated to Non-Alcoholic Fatty Liver Disease in HIV Mono-Infected Patients"

_jpm, 2022, doi:10.3390/jpm12020282_

Round 1
Reviewer 1 Report
The authors have investigated if using 5 simple biomarkers for detection of NAFLD in PWH patients could save important resources in HIV clinics. The study is well conducted and well written.
- The authors should elaborate on why they used these 5 simple fibrosis biomarkers in their study. What data suggested to them to use for example AST:ALT ratio <0.8 to be a biomarker? This needs to be elaborated in the introduction.
- Diet or BMI plays an important role in NAFLD. Is there any data to indicate dietary choices of the participants involved in the study? If not, can authors comment on how diet would have influenced BMI and hence results obtained particularly in PWH taking anti virals?
Reviewer 2 Report
Sebastiani et al described the role of serum biomarkers in predicting significant liver fibrosis in HIV positive subjects with no alcohol abuse or viral coinfection. The availability of unexpensive tools for diagnosing and monitoring over time hepatic disease is certainly useful but their role in evaluation of PWH in a high income country is limited and TE should be performed : conversely, it is possible that the workflow could help in optimizing costs in middle and low income countries.
Point 1. HIV patients included in the study had elevated CD4+ cell count and most were on successful ART. Previous works (i.e. Perazzo et al, J Int AIDS Soc. 2018;21:e25201) showed that a CD4+ count <200 cells/mm3 was associated with fibrosis: how many subjects with low CD4+ count ( < 200 and < 350 cells/mm3) were included in your study population ? Please discuss the finding.
Point 2. lines 232-235 “Since APRI and FIB-4 yielded the highest decrease in TE referrals, we conducted a multivariable analysis of independent predictors of discordance between low APRI (<0.5) 233 or FIB-4 (<1.3) and high TE (>7.1). After adjustments, high BMI and triglycerides were independent predictors of discordance for APRI and FIB-4”: high BMI and triglycerides are two parameters that may change many times over time and it can be difficult to rely on for a longitudinal monitoring. Please discuss.
Round 2
Reviewer 1 Report
The authors have addressed my comments. I don't have any more comments and support publication at this time.
Reviewer 2 Report
The authors successfully addressed all the points.